# *In vitro* modeling of *Batrachochytrium dendrobatidis* infection of the amphibian skin

Elin Verbrugghe[1]*, Pascale Van Rooij[1], Herman Favoreel[2], An Martel[1☯], Frank Pasmans[1☯]

1 Department of Pathology, Bacteriology and Avian Diseases, Faculty of Veterinary Medicine, Ghent University, Merelbeke, Belgium, 2 Department of Virology, Parasitology and Immunology, Faculty of Veterinary Medicine, Ghent University, Merelbeke, Belgium

☯ These authors contributed equally to this work.
* elin.verbrugghe@ugent.be

**Data Availability Statement:** All data underlying the results are presented in the paper and its Supporting Information files.

**Funding:** E. V. was supported by the Research Foundation Flanders (FWO grants 12E6616N and

## Abstract

The largest current disease-induced loss of vertebrate biodiversity is due to chytridiomycosis and despite the increasing understanding of the pathogenesis, knowledge unravelling the early host-pathogen interactions remains limited. *Batrachochytrium dendrobatidis* (*Bd*) zoospores attach to and invade the amphibian epidermis, with subsequent invasive growth in the host skin. Availability of an *in vitro* assay would facilitate in depth study of this interaction while reducing the number of experimental animals needed. We describe a fluorescent cell-based *in vitro* infection model that reproduces host-*Bd* interactions. Using primary keratinocytes from *Litoria caerulea* and the epithelial cell line A6 from *Xenopus laevis*, we reproduced different stages of host cell infection and intracellular growth of *Bd*, resulting in host cell death, a key event in chytridiomycosis. The presented *in vitro* models may facilitate future mechanistic studies of host susceptibility and pathogen virulence.

## Introduction

Chytridiomycosis plays an unprecedented role in the currently ongoing sixth mass extinction [1]. Worldwide, this fungal disease has caused catastrophic amphibian die-offs and it is considered as one of the worst infectious diseases among vertebrates in recorded history [1–3]. Two chytrid species, *Batrachochytrium dendrobatidis* (*Bd*) [4] and *Batrachochytrium salamandrivorans* (*Bsal*) [5], have been identified as the etiological agents of chytridiomycosis. Both pathogens parasitize amphibians by colonizing the keratinized layers (*stratum corneum*), resulting in disturbance of skin functioning and possibly leading to death in these animals [4, 6–10]. Whereas *Bsal* induces the formation of skin ulcera [5], *Bd* typically induces epidermal hyperplasia, hyperkeratosis and increased sloughing rates, eventually leading to the loss of physiological homeostasis (low electrolyte levels) [11–18]. The worldwide distribution of chytridiomycosis, its rapid spread, high virulence, and its remarkably broad amphibian host range lead to considerable losses in amphibian biodiversity [1].

*Bd*-induced chytridiomycosis was first described 20 years ago [4] and several studies have documented *Bd* growth and development at morphological and ultrastructural levels [19–21].

1507119N). Financial support of P. V. R. is funded by the Ghent University Special Research Fund (BOF13/PDO/130). The funders had no role in study design, data collection and analysis, decision to publish, or preparation of the manuscript.

**Competing interests:** The authors have declared that no competing interests exist.

The general *Bd*-infection steps have been described as attraction to a suitable host, attachment of zoospores to the host skin, zoospore germination, germ tube development and penetration into the skin cells, leading to endobiotic growth of this pathogen inside host cells which eventually results in the loss of host cytoplasm [20]. Despite recent advances in understanding the pathogenesis, fundamental knowledge about the early infection process at a cellular level, crucial in understanding disease pathogenesis, is however still limited [6–11, 21–22].

Infectious diseases are commonly studied *in vitro* by assessing the interaction of a pathogen with host cells. This is a reductionist approach, but one that can advance the understanding of mechanisms that underlie infection and disease. After two decades of chytrid research, a cell-based assay is lacking and the focus still remains on *in vivo* experimentation. To date, infectivity and the pathogenicity of *Bd* have mostly been studied using light microscopy (LM), scanning electron microscopy (SEM) and transmission electron microscopy (TEM) on *in vivo*-infected skin tissues or *ex vivo*-infected skin explants [20, 23]. The main objective of the current study was to establish a cell-based assay that mimics the colonization stages of *Bd in vitro*, allowing rapid and efficient screening of host-*Bd* interactions. We first optimized an early-infection model showing attachment of *Bd* to primary amphibian keratinocytes (PAK), followed by internalization of *Bd* in these host cells. Secondly, we developed an invasion model using the *Xenopus laevis* kidney epithelial cell line A6 mimicking the complete *Bd* colonization cycle *in vitro*.

## Materials and methods

### Experimental animals

We isolated PAK from adult *Litoria caerulea* (captive bred). Upon arrival and before the start-up of the experiments we examined kin swabs for the presence of *Bd* by quantitative PCR (qPCR) [24]. Husbandry and euthanasia methods were in accordance with the guidelines of the Ethical committee of the Faculty of Veterinary Medicine (Ghent University). Animals were euthanized by intracoelomic injection of sodium pentobarbital (Annex IV of the EU directive 2010/63). For the isolation of primary keratinocytes, ethical permission by the ethical committee of the Faculty of Veterinary Medicine (Ghent University) was not required under Belgian and European legislation (EU directive 2010/63/EU).

### *Batrachochytrium dendrobatidis* growth conditions

We carried out the inoculations with *Bd* strain JEL 423. This strain was isolated from an infected *Phyllomedusa lemur* frog in Panama and is a representative of the *Bd* global panzootic lineage [25]. The *Bd* strain was routinely cultured in TGhL broth (1.6% tryptone, 0.4% gelatin hydrolysate and 0.2% lactose in $H_2O$) in 75 cm$^2$ cell culture flasks at 20˚C for 5 days. We collected the *Bd* spores from a full-grown culture containing mature sporangia. Once the zoospores were released, the medium containing the zoospores was collected and passed over a sterile mesh filter with pore size 10 μm (PluriSelect, Leipzig, Germany). We used the flow-through as the zoospore fraction (> 90% purity).

### Cell culture: Isolation of PAK

Isolation of PAK from *Litoria caerulea* frogs was performed as previously described [20,23], with minor modifications. In brief, after euthanizing the frogs, we washed them in plastic containers containing respectively 70% ethanol, 70% Leibovitz L-15 medium without phenol red (3 times) (Fisher Scientific, Aalst, Belgium), $Ca^{2+}/Mg^{2+}$-free Barth's solution (CMFB; Bilaney Consultants GmbH, Düsseldorf, Germany), 1.25 mM ethylenediaminetetraacetic acid (EDTA; Sigma-Aldrich, Overijse, Belgium) in CMFB for 5 min and 70% L-15 medium (twice) at 4˚C.

Next, we excised ventral skin, which we rinsed at apical and basal side with 70% L-15 medium. From each donor animal a skin sample was taken, fixed in 70% EtOH and tested for the presence of *Bd* by qPCR [24]. We then cut the skin into 10 x 20 mm wide strips, which were incubated overnight in MatriSperse™ Cell Recovery Solution (BD Biosciences, Massachusetts, USA) at 4˚C. Subsequently, we peeled off the the cornified skin layers using sterile needles and forceps. To obtain single cell suspension, we incubated the cornified skin in 10 U/ml dispase solution (Fisher scientific) in 70% L-15 medium at 20˚C, 5% $CO_2$. Finally the cells were suspended by repetitive pipetting, washed in 70% L-15 medium and resuspended in the appropriate cell culture medium for invasion assays.

## Cell culture: Continuous A6 cell line

The *Xenopus laevis* kidney epithelial cell line A6 (ATCC-CCL 102) was grown in 75 cm$^2$ cell culture flasks and maintained in complete growth medium (74% NCTC 109 medium (Fisher Scientific), 15% distilled water, 10% fetal bovine serum (FBS) and 1% of a 10 000 U/ml penicillin-streptomycin solution (Fisher Scientific)) and the cells were incubated at 26˚C and 5% $CO_2$ until they reached confluence [26]. Using trypsin, we detached the cells, washed them with 70% Hanks' Balanced Salt Solution without $Ca^{2+}$, $Mg^{2+}$ (HBSS-; Fisher Scientific) by centrifugation for 5 min at 1500 rpm and resuspended them in the appropriate cell culture medium for invasion assays.

## Fluorescent *in vitro* model to assess adhesion and invasion of *Bd* in PAK

PAK are only usable for 1 to 4 days and the lifecycle of *Bd* takes approximately 4 to 5 days [19]. As such, these cells are not appropriate to examine the complete maturation process of this pathogen, but they can be used to investigate the early steps in *Bd*-host cell interaction, including adhesion and invasion. To visualize these early pathogen interactions (4 hours: adhesion and 24 hours: invasion), we stained the PAK with 3 μM CellTracker™ Green CMFDA (Fisher Scientific) according to the manufacturer's guidelines. After centrifugation for 5 min at 1500 rpm, we suspended the cells in cell medium A (70% L-15 medium, 20% distilled water and 10% FBS) and seeded 10$^5$ cells per well in 24-well tissue culture plates containing collagen-coated glass coverslips. PAK were allowed to attach for 1 hour at 20˚C and 5% $CO_2$ after which they were washed with 70% Hanks' Balanced Salt Solution with $Ca^{2+}$, $Mg^{2+}$ (HBSS+; Fisher Scientific). Next, we inoculated the cells with *Bd* zoospores stained with 3 μM CellTracker™ Red CMTPX (Fisher Scientific) [27] in cell medium B (40% L-15 medium, 55% distilled water and 5% FBS), to ensure the mobility of the zoospores, at a multiplicity of infection (MOI) of 1:10. After 2 hours, we gently washed the infected cells three times with 70% HBSS+ to remove non-adherent spores and we replaced cell medium B by cell medium A. To asses early *Bd*-PAK interactions, the infected cells were washed three times and fixed in 0.5 ml 3.0% paraformaldehyde for 10 min at 4 hours and 24 hours post infection (p.i.). Finally, we used Hoechst (Fisher Scientific) for nuclear staining and we mounted the coverslips using ProLong Gold antifade mountant (Fisher Scientific). For visual confirmation of *Bd*-PAK interactions, we studied the cells using fluorescence microscopy and confocal laser scanning microscopy (CLSM), using appropriate filter sets. Detailed protocols are available at protocols.io (dx.doi.org/10.17504/protocols.io.8ihhub6).

## Fluorescent *in vitro* model to assess adhesion of *Bd* to A6 cells

To study *Bd* adhesion (less than 24 hours p.i.) in the epithelial cell line A6, the protocol described in PAK cells was slightly modified. After staining the A6 cells with 3 μM CellTracker™ Green CMFDA, we seeded 10$^5$ cells per well in 24-well tissue culture plates containing collagen-coated glass coverslips and the cells were allowed to attach for 2 hours at 20˚C and 5% $CO_2$. We then washed the cells three times with 70% HBSS+, after which we inoculated

them with *Bd* zoospores in cell medium B, at a MOI of 1:10. Two hours p.i., the cells were washed three times with 70% HBSS+ and we replaced cell medium B by cell medium A. Four hours p.i. the infected cells were washed three times with HBSS+ and we incubated them with Calcofluor White stain (10 μg/ml in 70% HBSS+; Sigma-Aldrich) for 10 min. Next, we washed the cells three times with 70% HBSS+, followed by fixation. Finally the cells were mounted and we analyzed them using fluorescence microscopy. Detailed protocols are available at protocols. io (dx.doi.org/10.17504/protocols.io.8thhwj6).

## Fluorescent *in vitro* model to assess invasion and intracellular maturation of *Bd* in A6 cells

To assess *Bd*-A6 cell interactions starting from 24 hours p.i., we seeded unstained A6 cells which we inoculated with unstained *Bd* zoospores as described above. At different time points (from 1 to 6 days p.i.), the *Bd*-A6 cell interactions were visualised as follows: infected cells were stained with 3 μM CellTracker™ Green CMFDA, washed three times with 70% HBSS+ and they were incubated with Calcofluor White stain (10 μg/ml in 70% HBSS+) for 10 min. After washing 3 times with HBSS+, we fixed the infected cells, permeabilized them for 2 min with 0.1% Triton® X-100 and incubated them for 60 min with a polyclonal antibody against *Bd* raised in rabbit (1/1000) [28]. After washing three times with 70% HBSS+, we incubated the samples with a monoclonal goat anti-rabbit Alexa Fluor 568 (1/500) antibody (Fisher Scientific; A11011). After an incubation of 1 hour, we washed the samples three times with 70% HBSS+, mounted them and finally analyzed them using fluorescence microscopy and CLSM. The Alexa Fluor 568 targeting *Bd* and Calcofluor White stainings were used in concert to assess the ability of *Bd* to penetrate the host cell. Calcofluor White is not internalized by A6 cells, whereas the Alexa Fluor 568 staining (targeting *Bd*) was applied after permeabilization of the host cells. As such, intracellular *Bd* will only be targeted by the Alexa Fluor 568 whereas extracellular *Bd* bodies will be bound by both the Alexa Fluor 568 and Calcofluor White stain. We included sham-infected cells as a negative control to check the cell morphology over different time points (S1 File). Detailed protocols are available at protocols.io (dx.doi.org/10.17504/protocols.io.8ishuee).

## Fluorescent caspase-3 staining to assess induction of apoptosis in A6 cells

To visualize the induction of cell death in *Bd*-infected A6 cells, we performed a fluorescent caspase-3 staining. Therefore, we seeded unstained A6 cells which were inoculated with unstained *Bd* zoospores as described above. At different time points (day 4 to 6 p.i.), the infected cells were fixed, permeabilized and incubated for 60 min with anti-caspase-3 primary antibody raised in rabbit (Sigma-Aldrich; C8487) 1/1000 diluted. After washing 3 times with 70% HBSS +, we treated the samples with goat anti-rabbit Alexa Fluor 568 (1/500) for 1 hour. We then washed the cells and treated them with Hoechst for 15 min. Finally, the cells were washed three times with 70% HBSS+, mounted and analyzed using fluorescence microscopy. We included sham-infected cells as a negative control and staurosporin-treated A6 cells (1 μM, 24 hours; Sigma-Aldrich) as a positive control (S2 File). Detailed protocols are available at protocols.io (dx.doi.org/10.17504/protocols.io.8tihwke)

## Results

### PAK can be used to reproduce the early infection stages of *Bd*

We first optimized an *in vitro* model using PAK, that could be used in research unravelling factors that underpin early pathogenesis in primary cells. Therefore, an invasion experiment was performed with fluorescently-labelled PAK cells of *Litoria caerulea* and fluorescently-labelled

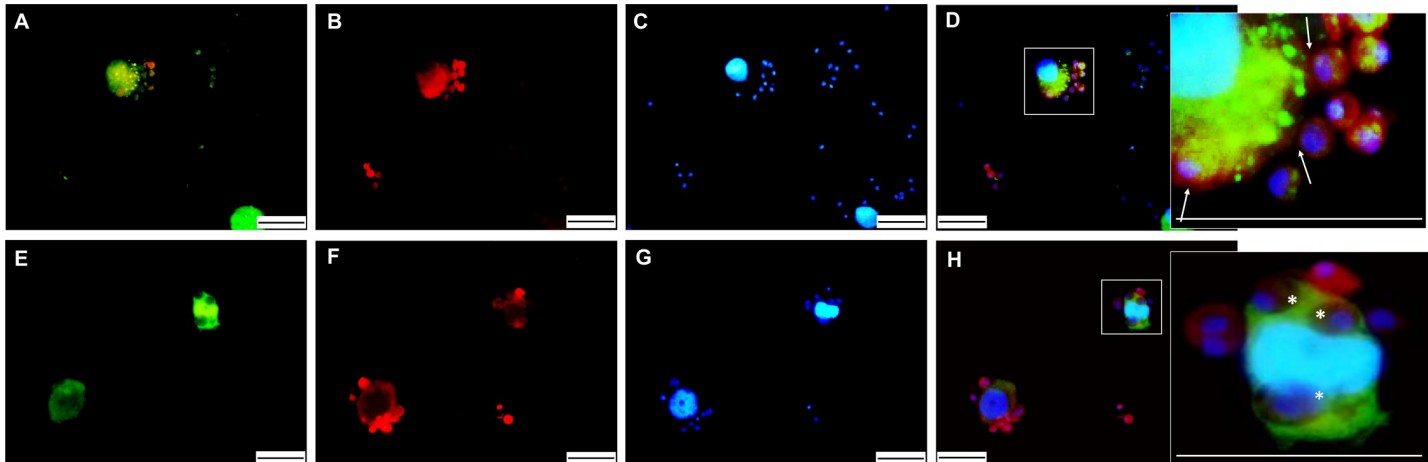

**Fig 1. Fluorescent staining of early stages of *Litoria caerulea* PAK infection by *Bd*. (A, E)** Host cells and **(B, F)** *Bd* spores were visualised using a green and red cell tracker, respectively. **(C, G)** Nuclear content was stained with Hoechst and all pictures were merged in **(D, H)**. **(A-D)** After 4 hours, initial contact was observed between host cells and *Bd* spores, as indicated by a white arrow **(D)**. **(E-H)** 24 hours after inoculation, marked intracellular colonization was seen in *Litoria caerulea* host cells, as indicated by a white asterisk **(H)**. Scale bar = 20 μm.

*Bd* spores that were incubated for 4 and 24 hours (Fig 1). After a 4-hour invasion period, clear contact between the spores and host cells was observed (Fig 1A–1D). When increasing the contact time to 24 hours, host cells seemed to be invaded by *Bd* spores as intracellular chytrid thalli were observed (Fig 1E–1H). To confirm this, confocal microscopy was used to determine the exact position of the chytrid thalli, showing a clear intracellular localization (Fig 2A–2C).

### The entire *Bd* colonization cycle can be mimicked using A6 cells

Fluorescent microscopy of PAK was shown to be useful to visualize the early host-pathogen steps, including attachment and invasion of *Bd*. We next tested whether similar results could be obtained working with the *Xenopus laevis* kidney epithelial cell line A6 (Figs 3 and 4). After an incubation of 4 hours, we observed the formation and growth of tubular structures, called germ tubes [20] (Fig 3A). Using a combination of Alexa Fluor 568 targeting *Bd* and a Calcofluor White staining allowed us to discriminate between the intracellular and extracellular localization of *Bd*. From 1 day p.i. on, the germ tubes penetrated the A6 cells (Fig 3B). After germ tube protrusion into the cells, both epibiotic and endobiotic *Bd* growth were observed. Epibiotic *Bd* growth was limited to *Bd* development outside the host cells, whereas endobiotic growth was characterized by intracellular *Bd* colonization (Fig 4). Both at 1 and 2 days p.i., an intracellular swelling was formed at the end of the germ tube, giving rise to a new *Bd* thallus (Fig 3C). As shown in Fig 3D, 2 to 3 days p.i. intracellular colonization was observed in A6 cells, which was also confirmed using CLMS (Fig 2D–2F). At day 3 and 4 p.i., maturation of the intracellular thalli was observed with the formation of large intracellular zoosporangia (Fig 3E). The formation of a discharge tube could be seen at day 4 to 5 p.i. (Fig 3F) through which *Bd* contents was released into the cell (Fig 3G), eventually leading to the induction of host cell death. From day 5 p.i. onward, marked caspase-3 activation was observed in *Bd*-associated A6 cells, a key event in apoptosis induction of host cells (Fig 3H).

### Discussion

Chytridiomycosis is increasingly recognized as a challenge for wildlife conservation. The power of a single disease to affect an entire vertebrate class and the fact that mitigation of *Bd*

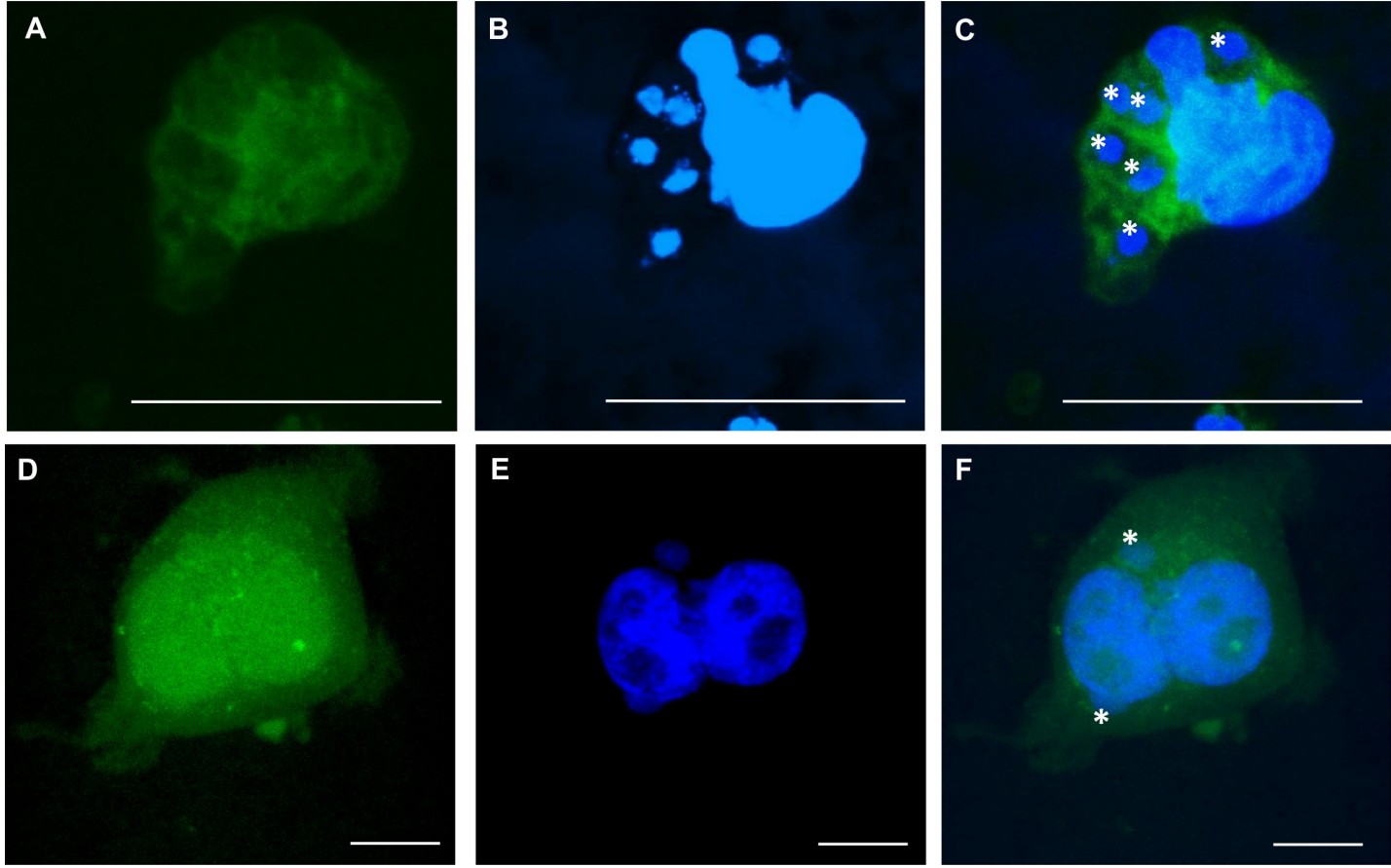

**Fig 2. Confocal microscopy of *Bd*-infected host cells after 24 hours.** Invasion of *Bd* inside **(A-C)** PAK of *Litoria caerulea* and **(D-F)** continuous A6 cells of *Xenopus laevis* was analyzed using confocal microscopy. **(A, D)** Host cells were stained with a green cell tracker and **(B, E)** nuclear content was stained with Hoechst. Both stainings were merged in **(C, F)**. By scanning different layers within the cell via confocal microscopy, *Bd* spores were validated being intracellular, as indicated by a white asterisk. Scale bar = 20 μm.

and *Bsal* in nature is still in its infancy [29], makes it one of the most destructive diseases ever recorded [1]. To date, a lot of research has focused on ecology and epidemiology of this fungal disease [1, 30–31] and although fundamental knowledge of the disease's pathogenesis is increasing, still knowledge gaps remain [32]. We here present *in vitro* infection models, intended to study *Bd*-host interactions in order to further explore the gaps in our understanding of chytridiomycosis. To date, *in vivo* experimentation still is the golden standard in *Bd* and *Bsal* research. To understand host-pathogen interactions in natural systems, researchers often turn to laboratory infection experiments. Although *in vivo* research has tremendous value for understanding disease processes, the availability of *in vitro* infection models could provide a first line tool to gain insight into host-pathogen interactions which will reduce the number of animals used in infection trials [33].

We showed that primary keratinocytes could be useful to mimic and examine the early *Bd*-host interactions, which until now have only been described using light microscopy and TEM of *Bd*-infected skin explants [20]. Previously, it has been stated that these cells cannot be used to study host-chytrid interactions because of the incompatibility of commonly-used culture media and the motility of *Bd* zoospores [23]. This obstacle was circumvented by diluting the cell culture medium during the first two hours of contact between *Bd* and the cells, guaranteeing the motility of the *Bd* spores during the adhesion process. Not all amphibian species are equally sensitive to

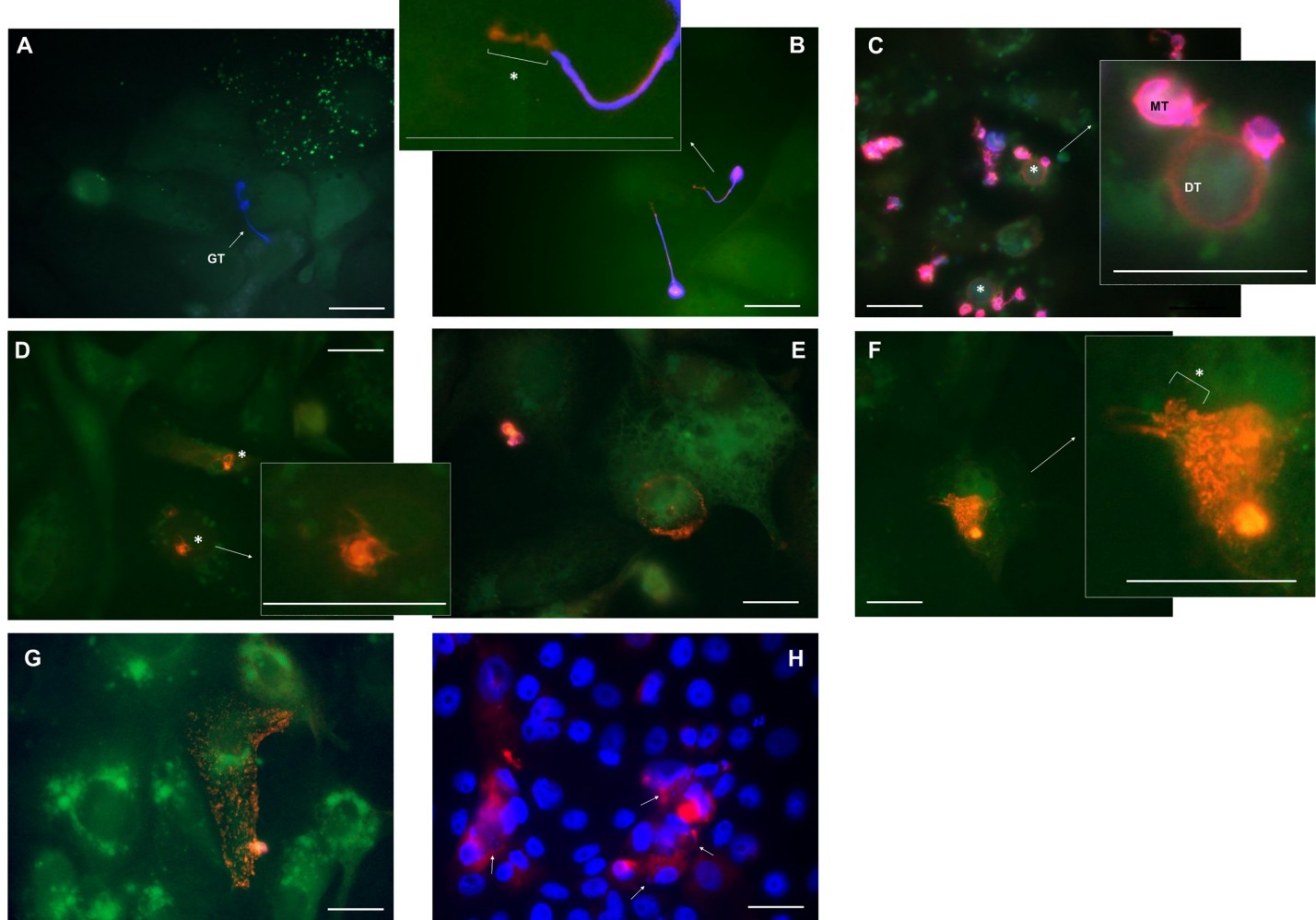

**Fig 3. *Bd* development in A6 cells.** Shown is an overlay of the fluorescent signals of **(A-G)** *Bd*-infected A6 cells (green cell tracker), extracellular *Bd* (Calcofluor White (blue)) and extra-and intracellular *Bd* (Alexa Fluor 568 (red)) or **(H)** caspase-3 activation (red) and nuclear content (Hoechst (blue)). **(A)** Four hours after inoculation, formation of germ tubes (GT) was observed and **(B)** within 24 hours, these tubular structures penetrated the A6 cells (*). **(C)** At day 1–2 p.i., new intracellular chytrid thalli (*) are formed and the cell content of the mother thallus (MT) is transferred into the new daughter thallus (DT). **(D)** At day 2–3 p.i., the emptied mother thallus evanesces, resulting in intracellular *Bd* bodies (*) that **(E)** develop intracellularly into sporangia at day 3–4 p.i. **(F)** Once the sporangia reach the stage of a mature zoosporangium (day 4–5 p.i.), they use a discharge tube (*) to release their contents into the A6 cells **(G)**. **(H)** At day 5–6 p.i., caspase-3 activation was observed in A6 cells associated with *Bd* (white arrow). Scale bar = 20 μm. Individual pictures of the different fluorescent channels can be found in S2 and S3 File.

chytridiomycosis and factors contributing to susceptibility of amphibians to this disease are not completely known [34–36]. However, specific attachment to a suitable host, induction of encystment and invasion of host cells are crucial and underexplored processes for successful colonization of this fungus. The described *in vitro* model may for example be used to look into adhesion factors or adherence mediators, that are possibly linked to host susceptibility.

Although working with primary cell cultures is more closely linked to the *in vivo* situation, cell lines provide the major advantage that they are standardized, immortalized and that no animals are needed. By using the epithelial cell line A6 from *Xenopus laevis* we were able to mimic the complete infection cycle of *Bd* and we showed that this model can be used to assess adhesion, invasion and maturation interactions, reflecting endobiotic development which is observed in susceptible amphibians [20]. Besides the intracellular colonization, we also observed epibiotic development of *Bd*, a type of growth which previously has been described

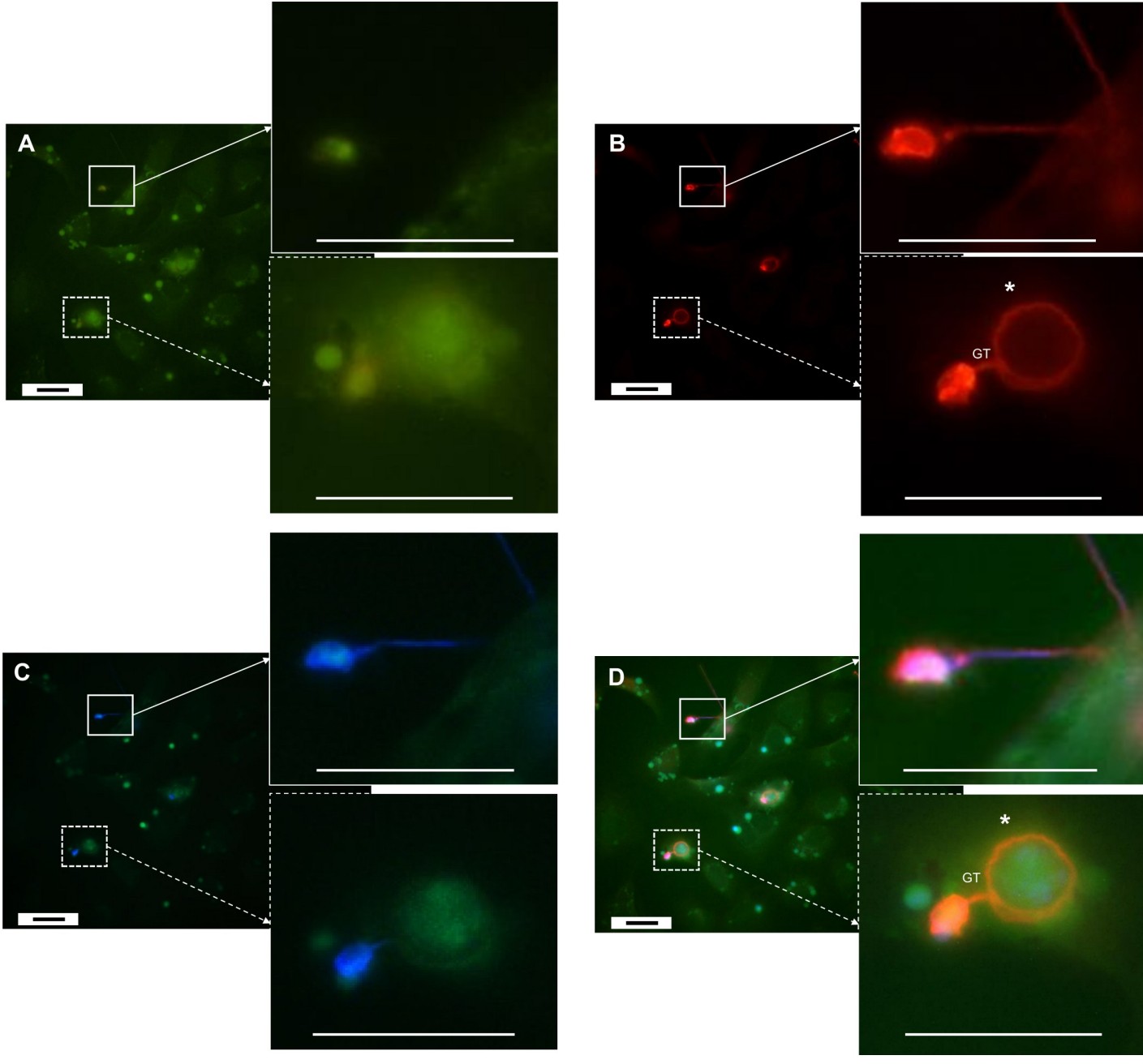

**Fig 4. Epibiotic and endobiotic growth of *Bd* on and in A6 cells, 2 days p.i.. (A)** *Bd*-exposed A6 cells were stained using a green cell tracker. **(B)** *Bd* was visualized using Alexa Fluor 568, resulting in red fluorescence of both intracellular and extracellular *Bd*. **(C)** The cell wall of extracellular *Bd* was coloured using Calcofluor White, showing blue fluorescence. The pictures were merged in **(D)**. Two days after inoculation, both epibiotic and endobiotic growth were observed. Epibiotic growth can be described as development outside the host cell (squares with a full line), which stains *Bd* both blue and red. Endobiotic growth (squares with a dashed line) is visualized as a red staining of the intracellular chytrid thalli (*) at the end of the germ tube (GT). Scale bar = 20 μm.

in infection trials with *ex vivo* skin explants of *Xenopus laevis* [20]. Up to date there is however no histological evidence of epibiotic growth of *Bd* occurring in nature. Therefore it could be suggested that the reported epibiotic growth is linked to the *in vitro*/*ex vivo* conditions, including the extracellular presence of nutrients from the cell culture medium, the lack of mucus and fungicidal skin secretions [37–42] and the lack of a normal skin microbiome [43–44].

In our *in vitro* model, apoptosis of A6 cells was observed when the zoospores were discharged into the cell by intracellular zoosporangia. As a result of this cell death the zoospores were released into the extracellular environment, ready to colonize new host cells, which (partly) deviates from the *in vivo* situation. In susceptible animals, germ tube-mediated invasion, establishment of intracellular thalli and spread of *Bd* to the deeper skin layers have been described, but this is followed by an upward migration by differentiating epidermal cells resulting in the releasement of the zoospores at the skin surface [19–21]. During *Bd*-induced chytridiomycosis, apoptosis has been reported as a key event, but the exact mechanism and role remains to be elucidated [45]. However, since epidermal cell death is positively associated with infection loads and morbidity [45], it is likely that cell death originates by colonization of many zoosporangia rather than the intracellular releasement of zoospores as observed in this *in vitro* model. Caution should always be exercised when extrapolating *in vitro* data to the *in vivo* situation, but *in vitro* cell culture models allow an experimental flexibility making them highly suitable to study host-pathogen interactions. Interestingly, the whole genome sequences of *Xenopus laevis* and *Bd* are known, permitting the study of transcriptional responses in host and pathogen during different infection stages. To date, different *Bd* lineages have been detected, all with their own virulence properties [25, 46–47]. The availability of an *in vitro* model using a continuous cell line may be used to analyze the differences in host-pathogen interactions between different *Bd* strains.

Summarized, for the first time we describe *in vitro* cell infection models that mimic *Bd* interactions with the amphibian skin ranging from adhesion, germ tube development, penetration into skin cells and invasive growth to the induction of host cell death. These *in vitro* models provide an import tool that may help understanding *Bd*-host interactions.

## Supporting information

**S1 File. Uninfected A6 cells.** Shown are the individual fluorescent signals of *Bd*-infected A6 cells (green cell tracker), extracellular *Bd* (Calcofluor White (blue)) and extra-and intracellular *Bd* (Alexa Fluor 568 (red)) of A6 cells at different time points after sham-infection (4 hours to 6 days). Scale bar = 20 μm.
(PDF)

**S2 File. Caspase-3 induction in *Bd*-infected A6 cells.** Shown are the individual fluorescent signals of caspase-3 activation (Alexa Fluor 568 (red)) and nuclear content (Hoechst (blue)), which were used in Fig 3. As a negative control sham-infected A6 cells were included and staurosporin-treated A6 cells (1 μM; 24 hours) served a positive control. Scale bar = 20 μm.
(PDF)

**S3 File. *Bd* development in A6 cells.** Shown are the individual fluorescent signals of *Bd*-infected A6 cells (green cell tracker), extracellular *Bd* (Calcofluor White (blue)) and extra-and intracellular *Bd* (Alexa Fluor 568 (red)) and their overlay pictures, which were used in Fig 3. Scale bar = 20 μm.
(PDF)

## Acknowledgments

The technical assistance Sarah Van Praet is greatly appreciated.

## Author Contributions

**Conceptualization:** Elin Verbrugghe, Pascale Van Rooij, An Martel, Frank Pasmans.

**Funding acquisition:** Elin Verbrugghe, An Martel, Frank Pasmans.

**Investigation:** Elin Verbrugghe.

**Methodology:** Elin Verbrugghe, Pascale Van Rooij, Herman Favoreel.

**Validation:** Elin Verbrugghe.

**Visualization:** Elin Verbrugghe, Pascale Van Rooij, Herman Favoreel.

**Writing – original draft:** Elin Verbrugghe.

**Writing – review & editing:** Elin Verbrugghe, Pascale Van Rooij, Herman Favoreel, An Martel, Frank Pasmans.

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
