## [Decision Letter · Decision Letter 0]

2 Oct 2019

PONE-D-19-18607

In vitro modeling of Batrachochytrium dendrobatidis infection of the amphibian skin

PLOS ONE

Dear Mrs Verbrugghe,

Thank you for submitting your manuscript to PLOS ONE. After careful consideration, we feel that it has merit but does not fully meet PLOS ONE’s publication criteria as it currently stands. Therefore, we invite you to submit a revised version of the manuscript that addresses the points raised during the review process.

Both reviewers enjoyed the manuscript and thought it appropriate for PLOS One.  Both also had some minor suggestions to clarify some points. Please address these points directly in the paper or in a reply where suitable.

We would appreciate receiving your revised manuscript by Nov 16 2019 11:59PM. To enhance the reproducibility of your results, we recommend that if applicable you deposit your laboratory protocols in protocols.io, where a protocol can be assigned its own identifier (DOI) such that it can be cited independently in the future. For instructions see: http://journals.plos.org/plosone/s/submission-guidelines#loc-laboratory-protocols

We look forward to receiving your revised manuscript.

Kind regards,

Jake Kerby, Ph.D.

Academic Editor

PLOS ONE

Journal Requirements:

Additional Editor Comments (if provided):

Both reviewers enjoyed the manuscript and thought it appropriate for PLOS One. Both also had some minor suggestions to clarify some points. Please address these points directly in the paper or in a reply where suitable.

Reviewers' comments:

Reviewer's Responses to Questions

**Comments to the Author**

1. Is the manuscript technically sound, and do the data support the conclusions?

Reviewer #1: Yes

Reviewer #2: Partly

2. Has the statistical analysis been performed appropriately and rigorously? 

Reviewer #1: Yes

Reviewer #2: N/A

3. Have the authors made all data underlying the findings in their manuscript fully available?

Reviewer #1: Yes

Reviewer #2: Yes

4. Is the manuscript presented in an intelligible fashion and written in standard English?

Reviewer #1: Yes

Reviewer #2: Yes

5. Review Comments to the Author

Reviewer #1: In this manuscript by Verbrugghe et al., the authors describe the development of in vitro methods for studying the pathogen Batrachochytrium dendrobatidis (Bd) and its interactions with amphibian hosts. Specifically, the investigators developed models with epithelial cell lines for in vitro experimentation with Bd. The found that they could successfully develop keratinocytes from Litoria caerulea and kidney epithelial cells from Xenopus laevis. As this is an extrememly important area of research, I believe this paper will make a valuable contribution to the literature. I cannot comment extensively on the methods for maintaining cell lines (for the skin and kidney epithelia) but the microscopy suggests that this approach will be very valuable (if it can be replicated) for understanding early pathogenesis of chytridiomycosis.

I have mostly minor comments:

Line 38- I believe stratum corneum should be in italics.

Line 40- Is the Bletz et al. reference appropriate here?

Line 43- Is this a paragraph break? If not, it should be.

Line 45- Change to "levels" (plural).

Lines 52-54 - I know a number of researchers that would dispute the notion that "surprisingly few studies have tackled this disease's pathogenesis". The authors make this comment at multiple points in the paper and it may bolster their argument to clarify that they are referring to molecular and cellular mechanisms of *early* pathogenesis. On this particular point (in line 52), it may make sense to include additional papers on pathogenesis of this disease (e.g., Carver et al. 2010, Marcum et al. 2010, Voyles et al. 2007)

Moreover, it appears there are a number of papers on pathogenesis and pathophysiology that the authors may have missed. As such, it may be worthwhile to review and include the following papers:

Bovo, R.P., Andrade, D.V., Toledo, L.F., Longo, A.V., Rodriguez, D., Haddad, C.F., Zamudio, K.R. and Becker, C.G., 2016. Physiological responses of Brazilian amphibians to an enzootic infection of the chytrid fungus Batrachochytrium dendrobatidis. Diseases of aquatic organisms, 117(3), pp.245-252.

Cramp, R.L., McPhee, R.K., Meyer, E.A., Ohmer, M.E. and Franklin, C.E., 2014. First line of defence: the role of sloughing in the regulation of cutaneous microbes in frogs. Conservation physiology.

Grogan, L.F., Skerratt, L.F., Berger, L., Cashins, S.D., Trengove, R.D. and Gummer, J.P., 2018. Chytridiomycosis causes catastrophic organism-wide metabolic dysregulation including profound failure of cellular energy pathways. Scientific reports, 8(1), p.8188.

Ohmer, M.E., Cramp, R.L., White, C.R. and Franklin, C.E., 2015. Skin sloughing rate increases with chytrid fungus infection load in a susceptible amphibian. Functional Ecology, 29(5), pp.674-682.

Russo, C.J., Ohmer, M.E., Cramp, R.L. and Franklin, C.E., 2018. A pathogenic skin fungus and sloughing exacerbate cutaneous water loss in amphibians. Journal of Experimental Biology, 221(9), p.jeb167445.

Wu, N.C., Cramp, R.L., Ohmer, M.E. and Franklin, C.E., 2019. Epidermal epidemic: unravelling the pathogenesis of chytridiomycosis. Journal of Experimental Biology, 222(2).

Young, S., Speare, R., Berger, L. and Skerratt, L.F., 2012. Chloramphenicol with fluid and electrolyte therapy cures terminally ill green tree frogs (Litoria caerulea) with chytridiomycosis. Journal of Zoo and Wildlife Medicine, 43(2), pp.330-337.

Line 58- This sentence is incomplete. It seems to be missing a verb.

Line 64- Perhaps it is semantics but I wonder if the term "colonization" is more appropriate than "infection" at various points of this paper?

Methods- A small point of preference - I recommend changing from passive voice to active voice throughout the methods.

Lines 128-132 For the audiences that will be reading this paper, I recommend providing references for these methods.

Lines 195-197- This sentence should be moved to the conclusions

Lines 261- I suggest changing "infections" to "diseases"

Lines 266- Change "chytrid" to "Bd and Bsal" because there are many chytrids that are not pathogens of amphibians.

Lines 275- I suggest changing to "This obstacle was circumvented...."

Lines 277- Same comment regarding the word "chytrid"

Reviewer #2: It is clear that this project required a lot of very meticulous work, so kudos to the authors! This new model will be a great resource for Bd research on interactions between Bd and host cells. I only had one concern regarding the limitations of the in vitro model, which doesn’t detract from the value of the model, but could be acknowledged more clearly.

Regarding the current study and references to the previous skin explant study, it seems that epibiotic growth may be related to the in vitro conditions, such as potentially some nutrients floating outside the cells in the culture medium and/or lack of a normal mucus layer that contains inhibitory bacteria and secretions. Thus, I am not sure that it is safe to assume that epibiotic growth occurs in nature (or at least occurs commonly), and this could influence conclusions made based on in vitro studies.

Another example of the in vitro conditions potentially influencing the life cycle of Bd in a way that could contrast with natural conditions could be the observation of cell death from zoosporangia discharging zoospores into the same host cell rather than to the cell surface. This could be from lack of normal cell layer orientation in vitro or another factor that is related to the in vitro conditions. While the observation of cell death from Bd makes sense, a more common mechanism in nature might be from colonization from many zoosporangia. Again, the in vitro conditions may be influencing the reinfection process in a way that does not fully replicate natural conditions.

Line 277: It might be useful to include the information about why PAK are useful for tracking the early infection process earlier in the paper, in the methods section, so the reader understands why these cells are not used to track the whole infection process.

Since PLOS ONE encourages authors to publish detailed protocols as supporting information, it seems that this would be appropriate for this paper, since the protocols are pretty complicated.

6. PLOS authors have the option to publish the peer review history of their article (what does this mean?). If published, this will include your full peer review and any attached files.

Reviewer #1: No

Reviewer #2: No

---

## [Author Response · Author response to Decision Letter 0]

29 Oct 2019

The authors would like to thank the editorial board and the reviewers for their valuable suggestions to improve our manuscript entitled “ In vitro modeling of Batrachochytrium dendrobatidis infection of the amphibian skin ”. Please find below the changes that were made according to the advice given by the reviewers and editor. The changes are marked with track changes in the final manuscript. 

We hope that the manuscript is now suitable for publication in Plos One in the present form, however we are willing to further modify our manuscript if needed.

Reviewer #1:

Line 38- I believe stratum corneum should be in italics.

This has been adapted.

Line 40- Is the Bletz et al. reference appropriate here?

This reference has been removed from line 40. 

Line 43- Is this a paragraph break? If not, it should be.

This indeed is a paragraph break and not a line break. We made this more clear in the article by adding more spacing between the different paragraphs.

Line 45- Change to "levels" (plural).

This has been adapted from “at a morphological and ultrastructural level’ to “at morphological and ultrastructural levels”. 

Lines 52-54 - I know a number of researchers that would dispute the notion that "surprisingly few studies have tackled this disease's pathogenesis". The authors make this comment at multiple points in the paper and it may bolster their argument to clarify that they are referring to molecular and cellular mechanisms of *early* pathogenesis. On this particular point (in line 52), it may make sense to include additional papers on pathogenesis of this disease (e.g., Carver et al. 2010, Marcum et al. 2010, Voyles et al. 2007)

- The authors added the references and reworded line 52 to: 

Despite recent advances in understanding the pathogenesis, fundamental knowledge about the early infection process at a cellular level, crucial in understanding disease pathogenesis, is however still limited (Berger et al., 2005b; Voyles et al., 2007: 2009; Carver et al., 2010; Marcum et al., 2010; Van Rooij et al., 2012; Greenspan et al., 2012; Brutyn et al., 2012; Fites et al., 2013).

- The first two lines in the abstract were also reworded to: 

The largest current disease-induced loss of vertebrate biodiversity is due to chytridiomycosis and despite the increasing understanding of the pathogenesis, knowledge unravelling the early host-pathogen interactions remains limited.

- The first paragraph of the discussion was reworded to: 

To date, a lot of research has focused on ecology and epidemiology of this fungal disease (Fisher et al., 2009; Kilpatrick et al., 2010; Scheele et al., 2019) and although fundamental knowledge of the disease’s pathogenesis is increasing, still knowledge gaps remain (Van Rooij et al., 2015).

It appears there are a number of papers on pathogenesis and pathophysiology that the authors may have missed. As such, it may be worthwhile to review and include the following papers:

The authors included the suggested references in the first paragraph of the introduction: 

Chytridiomycosis plays an unprecedented role in the currently ongoing sixth mass extinction (Scheele et al., 2019). Worldwide, this fungal disease has caused catastrophic amphibian die-offs and it is considered as one of the worst infectious diseases among vertebrates in recorded history (Lips 2016; Skerratt et al., 2007; Scheele et al., 2019). Two chytrid species, Batrachochytrium dendrobatidis (Bd) (Berger et al., 1998) and Batrachochytrium salamandrivorans (Bsal) (Martel et al., 2013), have been identified as the etiological agents of chytridiomycosis. Both pathogens parasitize amphibians by colonizing the keratinized layers (stratum corneum), resulting in disturbance of skin functioning and possibly leading to death in these animals (Berger et al., 1998; Voyles et al., 2007: 2009; Carver et al., 2010; Marcum et al., 2010; Brutyn et al., 2012). Whereas Bsal induces the formation of skin ulcera (Martel et al., 2013), Bd typically induces epidermal hyperplasia, hyperkeratosis and increased sloughing rates, eventually leading to the loss of physiological homeostasis (low electrolyte levels) (Berger et al., 2005b; Young et al., 2012; Cramp et al., 2014; Ohmer et al., 2015; Bovo et al., 2016; Grogan et al., 2018; Russo et al., 2018; Wu et al., 2019). The worldwide distribution of chytridiomycosis, its rapid spread, high virulence, and its remarkably broad amphibian host range lead to considerable losses in amphibian biodiversity (Scheele et al., 2019). 

Bovo RP, Andrade DV, Toledo LF, Longo AV, Rodriguez D, Haddad CF, et al. Physiological responses of Brazilian amphibians to an enzootic infection of the chytrid fungus Batrachochytrium dendrobatidis. Dis. Aquat. Organ. 2016; 117: 245-252.

Cramp RL, McPhee RK, Meyer EA, Ohmer ME, Franklin CE. First line of defence: the role of sloughing in the regulation of cutaneous microbes in frogs. Conserv. Physiol. 2014; 2: cou012.

Grogan LF, Skerratt LF, Berger L, Cashins SD, Trengove RD, Gummer JP. Chytridiomycosis causes catastrophic organism-wide metabolic dysregulation including profound failure of cellular energy pathways. Sci. Rep. 2018; 8: 8188.

Ohmer ME, Cramp RL, White CR, Franklin CE. Skin sloughing rate increases with chytrid fungus infection load in a susceptible amphibian. Funct. Ecol. 205; 29: 674-682.

Russo CJ, Ohmer ME, Cramp RL, Franklin CE. A pathogenic skin fungus and sloughing exacerbate cutaneous water loss in amphibians. J. Exp. Biol. 2018; 221: jeb167445.

Wu NC, Cramp RL, Ohmer ME, Franklin CE. Epidermal epidemic: unravelling the pathogenesis of chytridiomycosis. J. Exp. Biol. 2019; 222: jeb191817

Young S, Speare R, Berger L, Skerratt LF. Chloramphenicol with fluid and electrolyte therapy cures terminally ill green tree frogs (Litoria caerulea) with chytridiomycosis. J. Zoo Wildl. Med. 2012; 43: 330-337. 

Line 58- This sentence is incomplete. 

The sentence was changed from: “A reductionist approach, but one that can advance the understanding of mechanisms that underlie infection and disease.” To “This is a reductionist approach, but one that can advance the understanding of mechanisms that underlie infection and disease.”

Line 64- Perhaps it is semantics but I wonder if the term "colonization" is more appropriate than "infection" at various points of this paper?

The authors changed “infection” to “colonization” in: 

- The main objective of the current study was to establish a cell-based assay that mimics the colonization stages of Bd in vitro, allowing rapid and efficient screening of host-Bd interactions.

- Secondly, we developed an invasion model using the Xenopus laevis kidney epithelial cell line A6 mimicking the complete Bd colonization cycle in vitro.

- The entire Bd colonization cycle can be mimicked using A6 cells 

Methods- A small point of preference - I recommend changing from passive voice to active voice throughout the methods.

We changed the passive voice to active voice in the materials and methods section. This is indicated via track changes in the manuscript. 

Lines 128-132 For the audiences that will be reading this paper, I recommend providing references for these methods.

We added the reference Blooi et al., 2017 and we also uploaded the detailed protocols from the Materials and Methods section in the protocols.io database. 

Blooi M, Laking AE, Martel A, Haesebrouck F, Jocque M, Brown T, et al. Host niche may determine disease-driven extinction risk. PLoS ONE 2017; 12: e0181051.

Lines 195-197- This sentence should be moved to the conclusions

The authors moved this sentence to L366 –L368: 

We showed that primary keratinocytes could be useful to mimic and examine the early Bd-host interactions, which until now have only been described using light microscopy and TEM of Bd-infected skin explants (Van Rooij et al., 2012).

Lines 261- I suggest changing "infections" to "diseases"

This has been adapted. 

Lines 266- Change "chytrid" to "Bd and Bsal" because there are many chytrids that are not pathogens of amphibians.

This has been adapted. 

Lines 275- I suggest changing to "This obstacle was circumvented...."

This has been adapted. 

Lines 277- Same comment regarding the word "chytrid"

This has been adapted. 

Reviewer #2:

It is clear that this project required a lot of very meticulous work, so kudos to the authors! This new model will be a great resource for Bd research on interactions between Bd and host cells. I only had one concern regarding the limitations of the in vitro model, which doesn’t detract from the value of the model, but could be acknowledged more clearly. Regarding the current study and references to the previous skin explant study, it seems that epibiotic growth may be related to the in vitro conditions, such as potentially some nutrients floating outside the cells in the culture medium and/or lack of a normal mucus layer that contains inhibitory bacteria and secretions. Thus, I am not sure that it is safe to assume that epibiotic growth occurs in nature (or at least occurs commonly), and this could influence conclusions made based on in vitro studies.

The authors agree with the fact that epibiotic growth does not (commonly) occur in nature. Until now, the epibiotic growth has only has been described in ex vivo skin explants of Xenopus laevis. It has been hypothesized that if the growth of Bd remains epibiotic, then sloughing could be effective at removing encysted zoospores and resulting zoosporangia, explaining the tolerance of this species to Bd. However, up to date, this is only speculative and no scientific data have been published underlying this hypothesis. In fact, there is no conclusive histological evidence of how Bd manifests in this species under natural conditions. 

With this paper it is not our aim to report epibiotic growth as a phenomenon happening in nature, but it is a type of growth that occurs in vitro (possibly an artefact of nutritional conditions) and that the readers should be aware of. To provide more background information, we removed the following lines from the introduction: 

In ex vivo skin explants of Xenopus laevis, an epibiotic growth has been described where the fungus uses the epidermal cells as a nutrient source and develops upon the skin (Van Rooij et al., 2012). 

And the authors changed a part of the discussion to L386 –L397 : 

Although working with primary cell cultures is more closely linked to the in vivo situation, cell lines provide the major advantage that they are standardized, immortalized and that no animals are needed. By using the epithelial cell line A6 from Xenopus laevis we were able to mimic the complete infection cycle of Bd and we showed that this model can be used to assess adhesion, invasion and maturation interactions, reflecting endobiotic development which is observed in susceptible amphibians (Van rooij et al., 2012). Besides the intracellular colonization, we also observed epibiotic development of Bd, a type of growth which previously has been described in infection trials with ex vivo skin explants of Xenopus laevis (Van Rooij et al., 2012). Up to date there is however no histological evidence of epibiotic growth of Bd occurring in nature. Therefore it could be suggested that the reported epibiotic growth is linked to the in vitro/ex vivo conditions, including the extracellular presence of nutrients from the cell culture medium, the lack of mucus and fungicidal skin secretions (Pasmans et al., 2013; Ramsey et al. 2010; Rollins-Smith et al., 2009 2011; Smith et al., 2018; Woodhams et al., 2017) and the lack of a normal skin microbiome (Bates et al., 2018; Bletz et al., 2018). 

Another example of the in vitro conditions potentially influencing the life cycle of Bd in a way that could contrast with natural conditions could be the observation of cell death from zoosporangia discharging zoospores into the same host cell rather than to the cell surface. This could be from lack of normal cell layer orientation in vitro or another factor that is related to the in vitro conditions. While the observation of cell death from Bd makes sense, a more common mechanism in nature might be from colonization from many zoosporangia. Again, the in vitro conditions may be influencing the reinfection process in a way that does not fully replicate natural conditions.

In order to clarify the possible “in vitro” effects to the readers, we included the following lines to the discussion L398-L419: 

In our in vitro model, apoptosis of A6 cells was observed when the zoospores were discharged into the cell by intracellular zoosporangia. As a result of this cell death the zoospores were released into the extracellular environment, ready to colonize new host cells, which (partly) deviates from the in vivo situation. In susceptible animals, germ tube-mediated invasion, establishment of intracellular thalli and spread of Bd to the deeper skin layers has been described, but this is followed by an upward migration by differentiating epidermal cells resulting in the releasement of the zoospores at the skin surface (Van Rooij et al., 2012; Berger et al., 2005a; Greenspan et al., 2012). During Bd-induced chytridiomycosis apoptosis has been reported as a key event, but the exact mechanism and role remains to be elucidated (Brannely et al., 2017). However, since epidermal cell death is positively associated with infection loads and morbidity (Brannely et al., 2017), it is likely that cell death originates by colonization of many zoosporangia rather than the intracellular releasement of zoospores as observed in this in vitro model. Caution should always be exercised when extrapolating in vitro data to the in vivo situation, but in vitro cell culture models allow an experimental flexibility making them highly suitable to study host-pathogen interactions.

Line 277: It might be useful to include the information about why PAK are useful for tracking the early infection process earlier in the paper, in the methods section, so the reader understands why these cells are not used to track the whole infection process.

This part of the discussion was moved to the Materials and methods section as suggested by the reviewer: L151-L154: PAK are only usable for 1 to 4 days and the lifecycle of Bd takes approximately 4 to 5 days (Berger et al., 2005a). As such, these cells are not appropriate to examine the complete maturation process of this pathogen, but they can be used to investigate the early steps in Bd-host cell interaction, including adhesion and invasion.

Since PLOS ONE encourages authors to publish detailed protocols as supporting information, it seems that this would be appropriate for this paper, since the protocols are pretty complicated.

The authors agree that a detailed step by step protocol would be useful for the readers. We therefore uploaded the detailed protocols from the Materials and Methods section in the protocols.io database. This was mentioned in the article by providing the doi numbers.

---

## [Editor Report · Decision Letter 1]

31 Oct 2019

In vitro modeling of Batrachochytrium dendrobatidis infection of the amphibian skin

PONE-D-19-18607R1

Dear Dr. Verbrugghe,

We are pleased to inform you that your manuscript has been judged scientifically suitable for publication and will be formally accepted for publication once it complies with all outstanding technical requirements.

With kind regards,

Jake Kerby, Ph.D.

Academic Editor

PLOS ONE

Additional Editor Comments (optional):

Thanks for making the changes. These all look great!

---

## [Editor Report · Acceptance letter]

6 Nov 2019

PONE-D-19-18607R1 

*In vitro* modeling of *Batrachochytrium dendrobatidis* infection of the amphibian skin 

Dear Dr. Verbrugghe:

I am pleased to inform you that your manuscript has been deemed suitable for publication in PLOS ONE. Congratulations! Your manuscript is now with our production department. 

With kind regards,

on behalf of

Dr. Jake Kerby 

Academic Editor

PLOS ONE